# PHOTOSWAP:
# Personalized Subject Swapping in Images

**Jing Gu**[1]    **Yilin Wang**[2]    **Nanxuan Zhao**[2]    **Tsu-Jui Fu**[3]    **Wei Xiong**[2]    **Qing Liu**[2]
**Zhifei Zhang**[2]    **He Zhang**[2]    **Jianming Zhang**[2]    **HyunJoon Jung**[2]    **Xin Eric Wang**[1]*
[1]University of California, Santa Cruz
[2]Adobe   [3]University of California, Santa Barbara
https://photoswap.github.io/

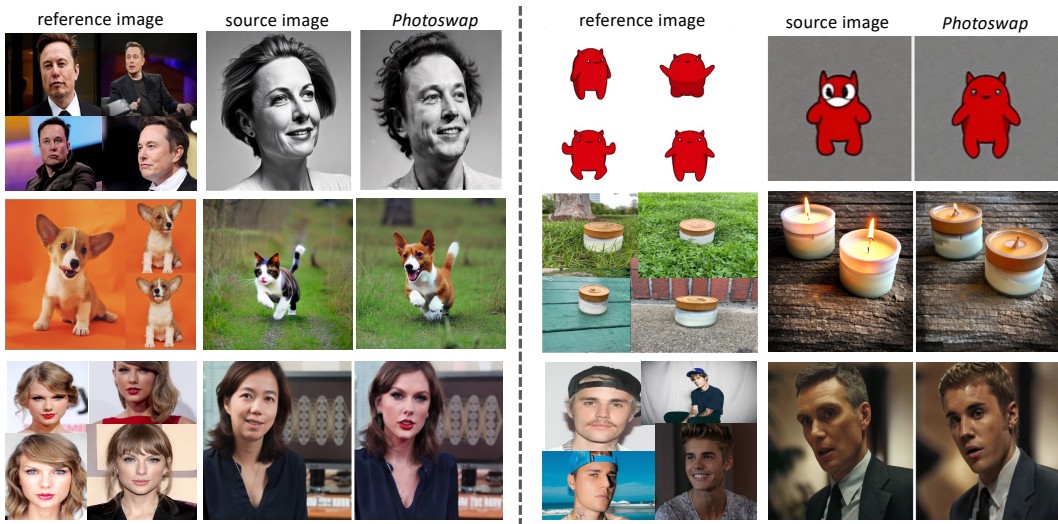

Figure 1: *Photoswap* can effortlessly replace the subject in a source image, which could be either synthetic (first two rows) or real (bottom row), with a personalized subject specified in reference images, while preserving the original subject pose and the composition of the source image.

## Abstract

In an era where images and visual content dominate our digital landscape, the ability to manipulate and personalize these images has become a necessity. Envision seamlessly substituting a tabby cat lounging on a sunlit window sill in a photograph with your own playful puppy, all while preserving the original charm and composition of the image. We present *Photoswap*, a novel approach that enables this immersive image editing experience through personalized subject swapping in existing images. *Photoswap* first learns the visual concept of the subject from reference images and then swaps it into the target image using pre-trained diffusion models in a training-free manner. We establish that a well-conceptualized visual subject can be seamlessly transferred to any image with appropriate self-attention and cross-attention manipulation, maintaining the pose of the swapped subject and the overall coherence of the image. Comprehensive experiments underscore the efficacy and controllability of *Photoswap* in personalized subject swapping. Furthermore, *Photoswap* significantly outperforms baseline methods in human ratings across subject swapping, background preservation, and overall quality, revealing its vast application potential, from entertainment to professional editing.

37th Conference on Neural Information Processing Systems (NeurIPS 2023).

# 1 Introduction

Imagine a digital world where the boundaries of reality and creativity blur, where a photograph of a tabby cat lounging on a sunlit window sill can effortlessly be transformed to feature your playful puppy in the same pose. Or envision yourself as a part of a famous movie scene, replaced seamlessly with the original character while preserving the very essence and composition of the scene. Can we achieve this level of personalized image editing, not just with expert-level photo manipulation skills, but in an automated, user-friendly manner? This question lies at the heart of *personalized subject swapping*, the challenging task of replacing the subject in an image with a user-specified subject, while maintaining the integrity of the original pose and composition. It opens up a plethora of applications in areas such as entertainment, advertising, and professional editing.

Personalized subject swapping is a complex undertaking that comes with its own set of challenges. The task requires a profound comprehension of the visual concept inherent to both the original subject and the replacement subject. Simultaneously, it demands the seamless integration of the new subject into the existing image. One of the critical objectives in subject swapping is to preserve the similar pose of the replacement subject. It is crucial that the swapped subject seamlessly fits into the original pose and scene, creating a natural and harmonious visual composition. This necessitates careful consideration of factors such as lighting conditions, perspective, and overall aesthetic coherence. By effectively blending the replacement subject with these elements, the final image maintains a sense of continuity and authenticity.

Existing image editing methods fall short in addressing these challenges. Many of these techniques are restricted to global editing and lack the finesse needed to seamlessly integrate new subjects into existing images. For example, for most text-to-image (T2I) models, a slightly prompt change could lead to a totally different image. Recent works (Nichol *et al.*, 2022; Meng *et al.*, 2022; Couairon *et al.*, 2022; Cao *et al.*, 2023; Zhang *et al.*, 2023) allow user to control the generation with an additional input such as user brush, semantic layout, or sketches. However, it is still challenging to guide the generation process to follow users' intent on the generation of object shape, texture, and identity. Other approaches (Hertz *et al.*, 2022; Tumanyan *et al.*, 2023; Mokady *et al.*, 2023) have explored the potential of using text prompts to edit image content in the context of synthetic image generation. Despite showing promise, these methods are not yet fully equipped to handle the intricate task of swapping subjects in existing images with user-specified subjects.

Therefore, we present *Photoswap*, a novel framework that leverages pre-trained diffusion models for personalized subject swapping in images. In our approach, the diffusion model learns to represent the concept of the subject ($O_t$). Then the representative attention map and attention output saved in the source image generation process will be transferred into the generation process of the target image to generate the new subject while keeping non-subject pixels unchanged. Our extensive experiments and evaluations demonstrate the effectiveness of *Photoswap*. Not only does our method enable the seamless swapping of subjects in images, but it also maintains the pose of the swapped subject and the overall coherence of the image. Remarkably, *Photoswap* outperforms baseline methods by a large margin in human evaluations of subject identity preservation, background preservation, and overall quality of the swapping (*e.g.*, 37.3% *vs.* 27.1% in terms of overall quality). The contributions of this work are as follows: **1)** We present a new framework for personalized subject swapping in images. **2)** We propose a training-free attention swapping method that governs the editing process. **3)** The efficacy of our proposed framework is demonstrated through extensive experiments including human evaluation.

# 2 Related Work

## 2.1 Text-to-Image Generation

In the early stages of text-based image generation, Generative Adversarial Networks (GANs) (Goodfellow *et al.*, 2020; Brock *et al.*, 2018; Karras *et al.*, 2019) were widely used due to their exceptional ability to produce high-quality images. These models aimed to align textual descriptions with synthesized images through multi-modal vision-language learning, achieving impressive results on specific domains (e.g., bird, chair and human face). When combined with CLIP (Radford *et al.*, 2021), a large pre-trained model that learns visual-textual representations from millions of caption-image pairs, GAN models (Crowson *et al.*, 2022) have demonstrated promising outcomes in cross-domain text-to-image (T2I) generation. Recently, T2I generation has seen remarkable progress with autoregressive (OpenAI, 2021; Ding *et al.*, 2021) and diffusion models (Nichol *et al.*, 2022; Gu *et al.*,

2022; OpenAI, 2022; Saharia *et al.*, 2022), offering diverse outcomes and can synthesize high-quality images closely aligned with textual descriptions in arbitrary domains.

Rather than focusing on T2I generation tasks without any constraints, subject-driven T2I generation (Nitzan *et al.*, 2022; Casanova *et al.*, 2021; Ruiz *et al.*, 2023) requires the model to identify the specific object from a set of visual examples and synthesize novel scenes incorporating them based on the input text prompts. Building upon modern diffusion techniques, recent approaches such as Dream-Booth (Ruiz *et al.*, 2023) and Textual Inversion (Gal *et al.*, 2023a,b; Kumari *et al.*, 2023; Mokady *et al.*, 2023) learn to invert special tokens from a given set of images. By combining these tokens with text prompts, they generate personalized unseen images. To improve data efficiency, retrieval augmentation techniques (Sheynin *et al.*, 2023; Blattmann *et al.*, 2022; Chen *et al.*, 2023) leverages external knowledge bases to overcome limitations posed by rare entities, resulting in visually relevant appearances and enhanced personalization. In our work, we aim to tackle personalized subject swapping, not only preserving the identity of subjects in reference images, but also maintaining the context of the source image.

### 2.2 Text-guided Image Editing

Text-guided image editing manipulates an existing image based on the input textual instructions, while preserving certain aspects or characteristics of the original image. Early works based on GAN models (Karras *et al.*, 2019) only limits to a certain object domain. Diffusion-based methods (Zhang *et al.*, 2023; Nichol *et al.*, 2022; Feng *et al.*, 2023) break this barrier and support text-guided image editing. Though these methods generate stunning results, many of them suffer from conducting local editing, and additional manual masks (Meng *et al.*, 2022; Zeng *et al.*, 2023; Meng *et al.*, 2022) are required to constrain the editing regions, which is often tedious to draw. By employing cross-attention (Hertz *et al.*, 2022) or spatial characteristics (Tumanyan *et al.*, 2023), the local editing can be achieved but struggles with non-rigid transformations (e.g., changing pose) and retaining the original image layout structure. While Imagic (Kawar *et al.*, 2023) addresses the need for non-rigid transformations by fine-tuning a pre-trained diffusion model to capture image-specific appearances, it requires test-time finetuning, which is not time-efficient for deployment. Moreover, relying solely on text as input lacks precise control. In contrast, we propose a novel training-free attention swapping scheme that enables precise personalization based on reference images, without the need for time-consuming finetuning.

### 2.3 Exemplar-guided Image Editing

Exemplar-guided image editing covers a broad range of applications, and most of the works (Wang *et al.*, 2019; Huang *et al.*, 2018; Zhou *et al.*, 2021) can be categorized as exemplar-based image translation tasks, conditioning on various information, such as stylized images (Liu *et al.*, 2021; Deng *et al.*, 2022; Zhang *et al.*, 2022), layouts (Yang *et al.*, 2023b; Li *et al.*, 2023c; Jahn *et al.*, 2021), skeletons (Li *et al.*, 2023c), sketches/edges (Seo *et al.*, 2023). With the convenience of stylized images, image style transfer (Liao *et al.*, 2017; Zhang *et al.*, 2020) receives extensive attention, replying on methods to build a dense correspondence between input and reference images, but it cannot deal with local editing. To achieve local editing with non-rigid transformation, conditions like bounding boxes and skeletons are introduced, but require drawing efforts from users, which sometimes are hard to obtain. A recent work (Yang *et al.*, 2023a) poses exemplar-guided image editing task as an inpainting task with the mask and transfers the semantic content from the reference image to the source one, with the context intact. Unlike these works, we propose a more user-friendly scenario by conducting personalized subject swapping with only reference images and obtain high-quality editing results. DreamEdit (Li *et al.*, 2023b) uses iterative inpainting to achieve subject replacement. Nevertheless, the existing approach fails to establish a comprehensive correlation between the source and target subjects. Conversely, our technique ensures that pivotal attributes, such as body gestures and facial expressions, remain unaltered.

## 3 Preliminary

Diffusion models are a type of generative model that operates probabilistically. In this process, an image is created by gradually eliminating noise from the target that is characterized by Gaussian noise. In the context of text-to-image generation, a diffusion model typically involves a process where an initial random image is gradually refined step by step, with each step guided by a learned model, until it becomes a realistic image. The changes to the image spread out and affect many pixels over

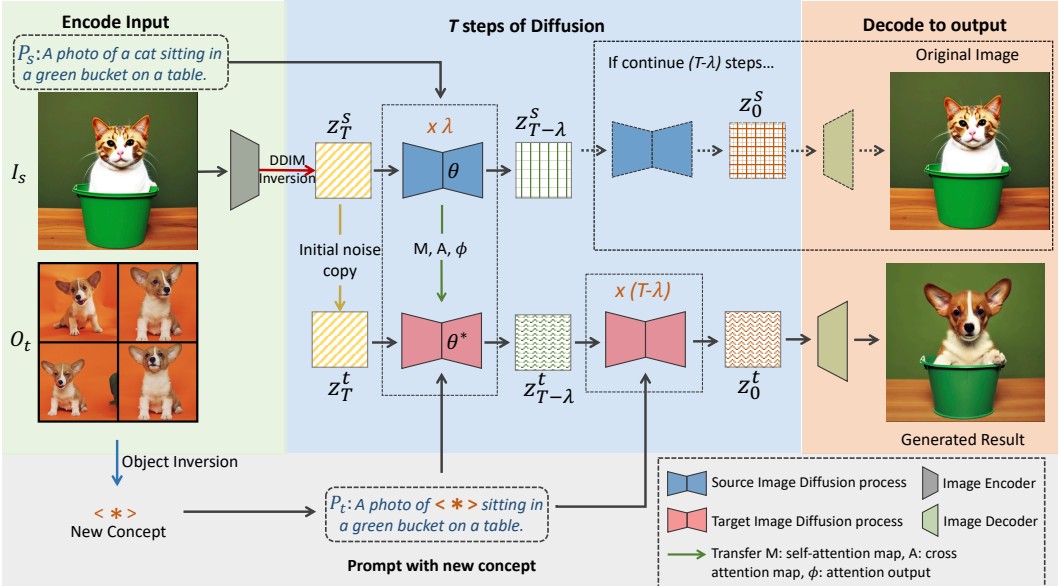

Figure 2: **The *Photoswap* framework.** Given several images of a new concept, the diffusion model first learns the concept and converts it into a token. The upper part is the generation process of the source image, while the bottom part is the generation process of target image. The initial noise feature $z_T^t$ is copied from $z_T^s$ of the source. The attention output and attention map in the source image generation process would be transferred to the target image generation process. The final feature $z_0^t$ is decoded to output the target image. Refer to Sec. 4 for more details.

time. Given an initial random noise $\boldsymbol{z}_T \sim \mathcal{N}(0, \mathbf{I})$, the diffusion model gradually denoise $\boldsymbol{z}_t$, which gives $\boldsymbol{z}_{t-1}$.

Diffusion models are probabilistic generative models that learn to generate images by simulating a random process called a diffusion process. In the image generation process, the diffusion model gradually predicts the noise at the current diffusion step and denoises to get the final image. In this study, we utilize a pre-trained text-to-image diffusion model, Stable Diffusion (Rombach *et al.*, 2022), which encodes the image into latent space and gradually denoises the latent variable to generate a new image. Stable Diffusion is based on a U-Net architecture (Ronneberger *et al.*, 2015), which generates latent variable $\boldsymbol{z}_{t-1}$ conditioned on a given text prompt $P$ and the latent variable $\boldsymbol{z}_t$ from the previous step $t$:

$$\boldsymbol{z}_{t-1} = \boldsymbol{\epsilon_\theta}(\boldsymbol{z}_t, P, t) \tag{1}$$

The U-Net consists of layers that include repetition of self-attention and cross-attention blocks. This study focuses on manipulating self-attention and cross-attention to achieve the task of personalized subject swapping.

## 4 The *Photoswap* Method

Providing a few reference images of a personalized target subject $O_t$, *Photoswap* can seamlessly swap it with another subject $O_s$ in a given source image $I_s$. The *Photoswap* pipeline is illustrated in Figure 2. To learn the visual concept of the target subject $O_t$, we fine-tune a diffusion model with reference images and do object inversion to represent $O_t$ using special token *. Then, to substitute the subject in the source image, we first obtain the noise $z_T$ [1] that can be used to reconstruct the source image $I_s$. Next, through the U-Net, we obtain the needed feature map and attention output in the self-attention and cross-attention layers, including $M$, $A$, and $\phi$ (which we will introduce in Sec. 4.2). Finally, during the target image generation process that is conditioned on the noise $z_T$ and the target text prompt $P_t$, in the first $\lambda$ steps, those intermediate variables ($M$, $A$, and $\phi$) would be replaced with corresponding ones obtained during the the source image generation process. In the last

---

[1]For a synthetic image, $z_T^*$ is the initial noise used to generate it. For a real image, we utilize an improved version of DDIM inversion (Song *et al.*, 2020) to get the initial noise and re-generate the source image. See Sec. 5.1 for details.

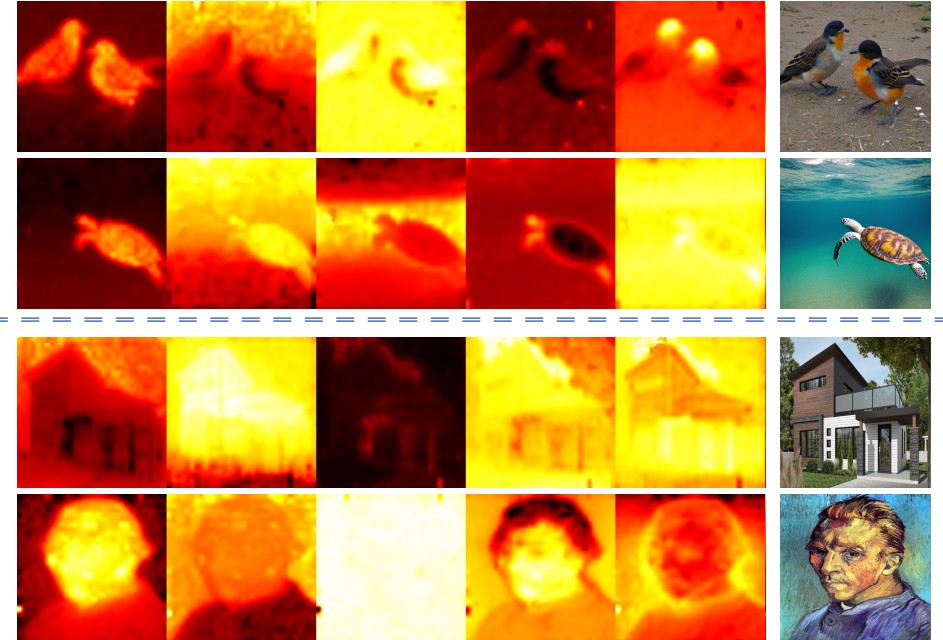

Figure 3: **SVD visualization of self-attention maps.** Each image's attention map is resized to 64x64 at every layer, and we calculate the average map across all layers for all diffusion time steps. Most significant components are extracted with SVD and visualized. Remarkably, the visualized results demonstrate a strong correlation with the layout of the generated image. The top two rows are visualization about synthetic images while the bottom two rows are about real images.

$(T - \lambda)$ steps, no attention swapping is needed and we can continue the denoising process as usual to obtain the final resulting image. Sec. 4.1 discusses the visual concept learning technique we used, and Sec. 4.2 details the training-free attention swapping method for controllable subject swapping.

## 4.1 Visual Concept Learning

Subject swapping requires a thorough understanding of the subject's identity and specific characteristics. This knowledge enables the creation of accurate representations that align with the source subject. The subject's identity influences the composition and perspective of the image, including its shape, proportions, and textures, which affect the overall arrangement of elements. However, existing diffusion models lack information about the target subject ($O_t$) in their weights because the training data for text-to-image generation models does not include personalized subjects. To overcome this limitation and generate visually consistent variations of subjects from a given reference set, we need to personalize text-to-image diffusion models accurately. Recent advancements have introduced various methods, such as fine-tuning the diffusion model with distinct tokens associated with specific subjects, to achieve this "personalization" (Gal *et al.*, 2023a; Ruiz *et al.*, 2023; Kumari *et al.*, 2023). In our experiments, we primarily utilize DreamBooth (Ruiz *et al.*, 2023) as a visual concept learning method. It's worth noting that alternative concept learning methods can also be effectively employed with our framework.

## 4.2 Controllable Subject Swapping via Training-free Attention Swapping

Subject swapping poses intriguing challenges, requiring the maintenance of the source image's spatial layout and geometry while integrating a new subject concept within the same pose. This necessitates preserving the critical features in the source latent variable, which encapsulates the source image information, and leveraging the influence of the target image text prompt $P_t$, which carries the concept token, to inject the new subject into the image.

The central role of the attention layer in orchestrating the generated image's layout has been well-established in prior works (Hertz *et al.*, 2022; Cao *et al.*, 2023; Tumanyan *et al.*, 2023). To keep non-subject pixels intact, we orchestrate the generation of the target image $I_t$ by transferring vital

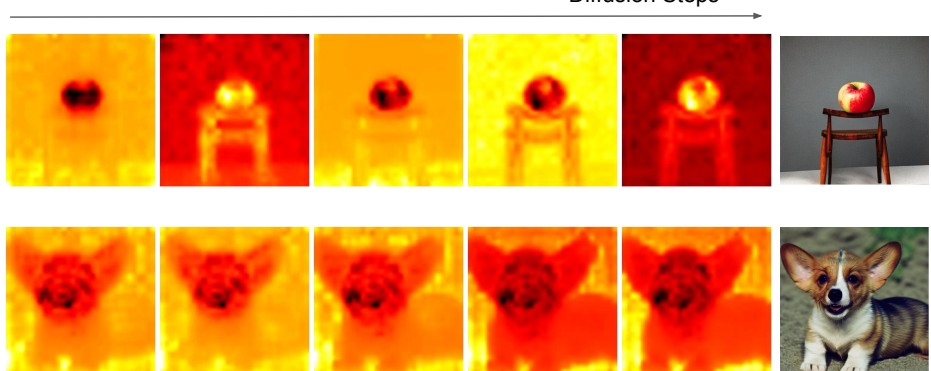

Figure 4: **Self-attention map visualization across diffusion time steps.** This representation reveals that the layout of the generated image is intrinsically embedded in the self-attention map from the initial steps. Consequently, to assert control over the layout, it is imperative to commence the attention swap at the earliest stages of the process.

variables to the target image generation process. Here, we explore how distinct intermediate variables within the attention layer can contribute to a controllable generation in the context of subject swapping.

Within the source image generation process, we denote the cross-attention map as $A_i^s$, the self-attention map as $M_i^s$, the cross-attention output as $\psi_i^s$, and the self-attention output as $\phi_i^s$. The corresponding variables in the target image generation process are denoted as $A_i^t$, $M_i^t$, $\psi_i^t$, $\phi_i^t$, where $i$ represents the current diffusion step.

In the self-attention block, the latent feature $z_i$ is projected into queries $q_i$, keys $k_i$, and values $v_i$. We obtain the self-attention block's output $\phi_i$ using the following equation:

$$\phi_i = M_i v_i \quad \text{where} \quad M_i = \text{Softmax}\left(q_i {k_i}^T\right) \tag{2}$$

where $M_i$ is the self-attention map, and $\phi_i$ is the feature output from the self-attention layer. The cross-attention block's output $\psi_i$ is:

$$\psi_i = A_i v_i \quad \text{where} \quad A_i = \text{Softmax}\left(q_i {k_i}^T\right) \tag{3}$$

where $A_i$ is the cross-attention map. In both self-attention and cross-attention, the attention map $M_i$ and $A_i$ are correlated to the similarity between $q_i$ and $k_i$, acting as weights that dictate the combination of information in $v_i$. In this work, the manipulation of the diffusion model focuses on self-attention and cross-attention within U-Net, specifically, swapping $\phi$, $M$, and $A$, while keeping $\psi$ unchanged.

**Self-attention map $M$**, as it calculates the similarity within spatial features after linear projection, plays a pivotal role in governing spatial content during the generation process. As visualized in Figure 3, we capture $M$ during the image generation and highlight the leading components via Singular Value Decomposition (SVD). This visualization reveals a high correlation between $M$ and the geometry and content of the generated image. Further, when visualizing the full steps of the diffusion process (Figure 4), we discern that the layout information is mirrored in the self-attention from the initial steps. This insight underscores the necessity of initiating the swap early on to prevent the emergence of a new, inherent layout.

**Cross-attention map $A$** is determined by both latent variable and text prompt, as in Equation 3, and $A_i^s v$ can be viewed as a weighted sum of the information from a text prompt. Copying $A_i^s$ to $A_i^t$ during the target image generation process improves the layout alignment between the source image and the target image.

**Self-attention output $\phi$**, derived from the self-attention layer, encapsulates rich content information from the source image, independent of direct computation with textual features. Hence, replacing $\phi_i^t$ with $\phi_i^s$ enhances the preservation of context and composition from the original image. Our observations indicate that $\phi$ exerts a more profound impact on the image layout than the cross-attention map $A$.

**Algorithm 1** The *Photoswap* Algorithm

---

**Inputs:** source image $I_s$, reference images $O_t$, source image text prompt $P_s$, target image text prompt $P_t$, diffusion model $\theta$

$\theta^* \leftarrow \theta, O_t \triangleright$ Finetune diffusion model to include the new concept
$\boldsymbol{z}_T^s \leftarrow DDIMInversion(ImageEncoder(I_s), P_s) \triangleright$ Using DDIM to guarantee re-construction
$\boldsymbol{z}_T^t \leftarrow \boldsymbol{z}_T^s \triangleright$ Using the same starting noise
**for** $i = T, T-1, ..., 1$ **do**
   $\epsilon^s, \phi_i^s, \boldsymbol{M}_i^s, \boldsymbol{A}_i^s \leftarrow \epsilon_{\theta*}(\boldsymbol{z}_i^s, P_s, i) \triangleright$ Denoise to get the attention output and map for source image
   $\phi_i^t, \boldsymbol{M}_i^t, \boldsymbol{A}_i^t \leftarrow \epsilon_{\theta*}(\boldsymbol{z}_i^t, P_t, i) \triangleright$ Denoise to get the attention output and map for target image
   $\phi_i^*, \boldsymbol{M}_i^*, \boldsymbol{A}_i^* \leftarrow \text{SWAP}(\phi_i^s, \boldsymbol{M}_i^s, \boldsymbol{A}_i^s, \phi_i^t, \boldsymbol{M}_i^t, \boldsymbol{A}_i^t, i)$
   $\epsilon^* \leftarrow \epsilon_{\theta*}(\boldsymbol{z}_i^t, P_t, i, \phi_i^*, \boldsymbol{M}_i^*, \boldsymbol{A}_i^*) \triangleright$ Denoise the updated attention map and output
   $\boldsymbol{z}_{i-1}^s \leftarrow DDIMSampler(\boldsymbol{z}_i^s, \epsilon^s) \triangleright$ Sample next latent variable for source image
   $\boldsymbol{z}_{i-1}^t \leftarrow DDIMSampler(\boldsymbol{z}_i^t, \epsilon^*) \triangleright$ Sample next latent variable for source image
**end for**
$I_t = ImageDecoder(\boldsymbol{z}_0^t)$
**return** $I_t$

**function** SWAP($\phi^s, \boldsymbol{M}^s, \boldsymbol{A}^s, \phi^t, \boldsymbol{M}^t, \boldsymbol{A}^t, i$)
   $\phi^* \leftarrow (i < \lambda_\phi)?\phi^s : \phi^t \triangleright$ Control self-attention feature swap
   $\boldsymbol{M}^* \leftarrow (i < \lambda_M)?\boldsymbol{M}^s : \boldsymbol{M}^t \triangleright$ Control self-attention Map swap
   $\boldsymbol{A}^* \leftarrow (i < \lambda_A)?\boldsymbol{A}^s : \boldsymbol{A}^t \triangleright$ Control cross-attention map swap
**return** $\phi^*, \boldsymbol{M}^*, \boldsymbol{A}^*$
**end function**

---

**Cross-attention output** $\psi$, emanating from the cross-attention layer, embodies the visual concept of the target subject. It is vital to note that substituting cross-attention output $\psi_i^s$ with $\psi_i^t$ would obliterate all information from the target text prompt $P_t$, as illustrated in Equation 3. Given that $k_i^t$ and $v_i^t$ are projections of target prompt embeddings, we retain $\psi_i^s$ unchanged to safeguard the target subject's identity.

Algorithm 1 provides the pseudo-code of our full *Photoswap* algorithm.

## 5 Experiments

### 5.1 Imlementation Details

For the implementation of subject swapping on real images, we require an additional process that utilizes an image inversion method, specifically the DDIM inversion (Song *et al.*, 2020), to transform the image into initial noise. This inversion method relies on a reversed sequence of sampling to achieve the desired inversion. However, there exist inherent challenges when this inversion process is applied in text-guided synthesis within a classifier-free guidance setting. Notably, the inversion can potentially amplify the accumulated error, which could ultimately lead to subpar reconstruction outcomes. To fortify the robustness of the DDIM inversion and to mitigate this issue, we further optimize the null text embedding, as detailed in Mokady *et al.* (2023). The incorporation of this optimization technique bolsters the effectiveness and reliability of the inversion process, consequently allowing for a more precise reconstruction. Without further notice, the DDIM inversion in this paper is enhanced by null text embedding optimization.

During inference, we utilize the DDIM sampling method with 50 denoising steps and classifier-free guidance of 7.5. The default step $\lambda_A$ for cross-attention map replacement is 20. The default step $\lambda_M$ for self-attention map replacement is 25, while the default step for self-attention feature $\lambda_\phi$ replacement is 10. Please refer to Appendix A for analysis on attention swapping step, and Appendix E for detailed Unet layer swapping. Note that the replacement steps may change to some specific checkpoint. As mentioned in Sec. 4, the target prompt $P_t$ is just source prompt $P_s$ with the object token being replaced with the new concept token. For concept learning, we mainly utilize DreamBooth (Ruiz *et al.*, 2023) to finetune a stable diffusion 2.1 to learn the new concept from 3 5 images. The learning rate is set to 1e-6. We use Adawm optimizer with 800 hundred training steps.

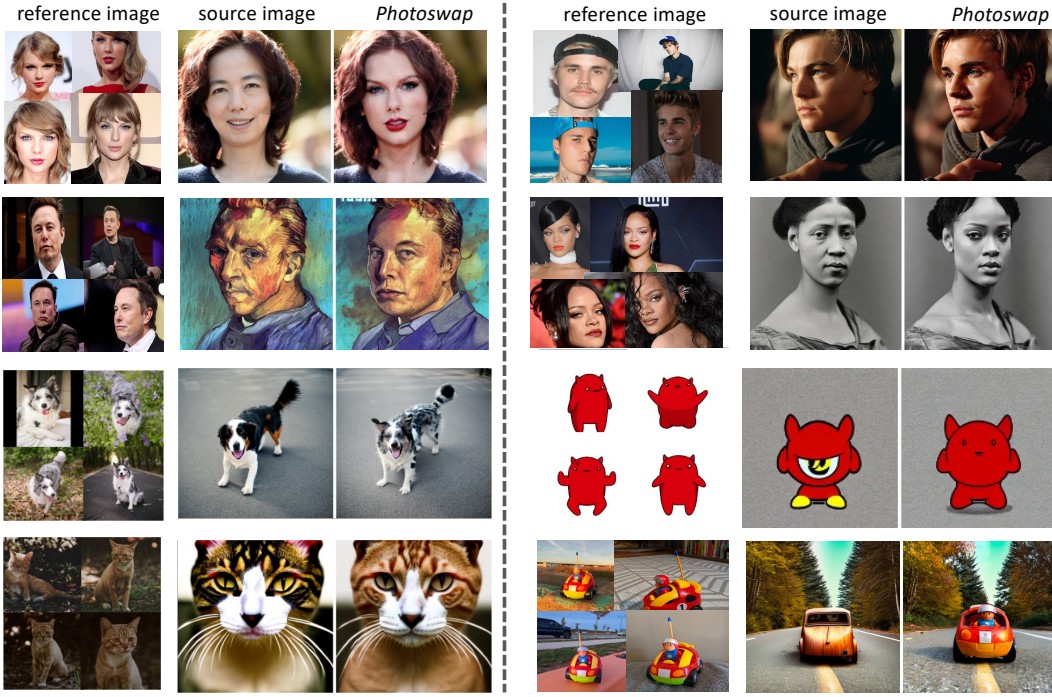

| reference image | source image | *Photoswap* | reference image | source image | *Photoswap* |

Figure 5: ***Photoswap* results across various object and image domains, demonstrating its wide applicability.** From everyday objects to cartoons, the diversity in subject swapping tasks has showcased the versatility and robustness of our framework across different contexts.

We finetune both the U-net and text encoder. The DreamBooth training takes around 10 minutes on a machine with 8 A100 GPU cards.

## 5.2 Personalized Subject Swapping Results

Figure 5 showcases the effectiveness of our *Photoswap* technique for subject swapping. Our approach excels at preserving crucial aspects such as spatial layout, geometry, and the pose of the original subject while seamlessly introducing a reference subject into the target image. Remarkably, even in cartoon images, our method ensures that the background remains intact during the subject change process. A notable example is the "cat" image, where our technique successfully retains all the intricate details from the source image, including the distinctive "Whiskers." This demonstrates our framework's ability to accurately capture and preserve fine-grained information during subject swapping. Please refer to Appendix C.

We further demonstrate the versatility of *Photoswap* by showcasing its effectiveness in multiple subject swap and occluded object swap scenarios. As depicted in Figure 6a, we present a source image featuring two sunglasses, which are successfully replaced with reference glass while preserving the original layout of the sunglasses. Similarly, in Figure 6b, we observe a source image with a dog partially occluded by a suit. The resulting swapped dog wears a suit that closely matches the occluded region. These examples serve to highlight the robustness of our proposed *Photoswap* method in handling various real-world cases, thereby enabling users to explore a broader range of editing possibilities. Refer Appendix F for identity control.

## 5.3 Comparison with Baseline Methods

Personalized object swap is a new task and there is no existing benchmark. However, we could modify the existing attention manipulation based methods. More specifically, we used the same concept learning method DreamBooth to finetune the same stable diffusion checkpoint to inject the new concept. To fairly compare with our results, we modified the existing prompt-based editing method P2P (Hertz *et al.*, 2022), an editing method based on diffusion models. Note that origin P2P only works on a pair of synthetic images, in our setting we use the same concept learning DreamBooth and fix the seed to allow concept swapping. On the other hand, PnP (Tumanyan *et al.*, 2023) could

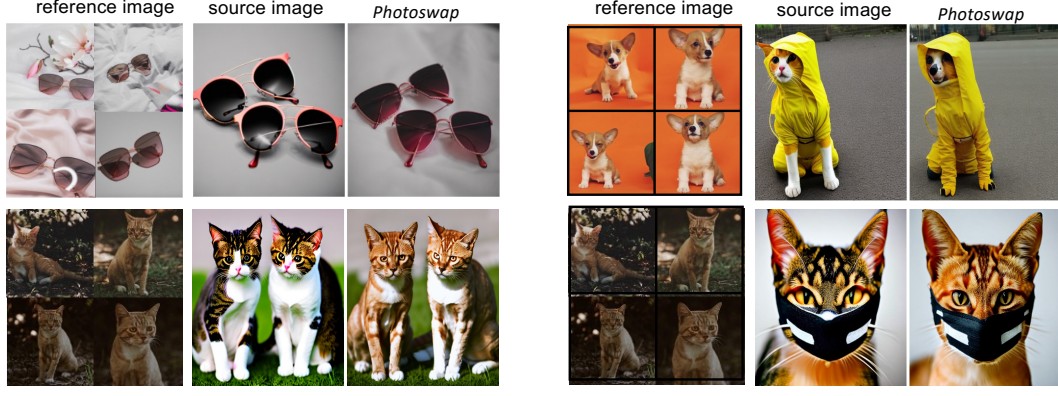

|              |                |
|:------------:|:--------------:|
| (a) Multi-subject swap. | (b) Occluded subject swap. |

Figure 6: ***Photoswap* results on multi-subject and occluded subject scenarios.** The results show that *Photoswap* can disentangle and replace multiple subjects at once. Also, *Photoswap* can identify the target object while avoiding influencing the non-subject pixels.

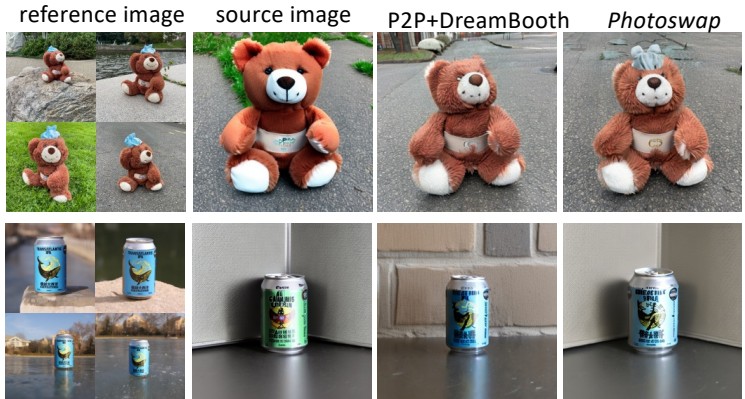

Figure 7: **Qualitative comparison between P2P+DreamBooth and *Photoswap*.** We can observe that P2P+DreamBooth is capable of achieving subject swapping. However, it faces challenges in preserving both the background and the reference subject accurately, while for *Photoswap*, it is robust to handle various cases.

also be implemented in a similar setting, however, we found PnP usually can not lead to satisfactory object swapping and may lead to a huge difference between the source image and the generated image. We suspect that it is because PnP is designed for image translation so it does not initiate the attention manipulation step from the beginning step. The qualitative comparison between *Photoswap* and P2P+DreamBooth is shown in Figure 7. Check Appendix B for performance on other concept learning method. We observe that P2P with DreamBooth could achieve basic object swap, but it still suffers from background mismatching issues.

**Human Evaluation.**    We conduct a human evaluation to study the editing quality by (1) *Which result better swaps the subject as the reference and keeps its identity*; (2) *Which result better preserves the background*; (3) *Which result has better overall subject-driven swapping*. We randomly sample examples and adopt Amazon MTurk[2] to compare between two results. Please refer to Appendix D for details. To avoid potential bias, we hired 3 Turkers for each sample. Table 1 demonstrates the comparison between our *Photoswap* and P2P. Firstly, more turkers (42.4%) denote that our *Photoswap* better swaps the subject yet keeps its identity at the same time. Moreover, we can also preserve the background in the source image (39.3% *vs.* 30.2%), which is another crucial goal of this editing. In summary, *Photoswap* precisely performs subject swapping and preserves the remaining part from the input, leading to an overall superiority (37.3%) to P2P (27.1%). Ku *et al.* (2023) shows *Photoswap*

---

[2]Amazon Mechanical Turk (MTurk): `https://www.mturk.com`.

Table 1: Qualitative comparison between *Photoswap* and P2P+DreamBooth, PnP+DreamBooth, MasaCtrl+DreamBooth separately. It includes 2000 comparisons on 400 image pairs, with 200 real images and 200 synthetic images. Each image pair contain 5 ratings from different human annotator in Amazon Turk. SS means Subject Swapping, BP means Background Preservation, and OQ means Overall Quality.

|  | *Photoswap* | P2P | Tie | *Photoswap* | PnP | Tie | *Photoswap* | MasaCtrl | Tie |
|---|---|---|---|---|---|---|---|---|---|
| SS | **42.4%** | 30.0% | 27.6% | **52.7%** | 22.1% | 25.2% | **79.1%** | 10.3% | 10.6% |
| BP | **39.3%** | 30.2% | 30.5% | **49.1%** | 20.7% | 30.2 % | **72.8%** | 10.2% | 17.0% |
| OQ | **37.3%** | 27.1% | 35.6% | **55.1%** | 22.4% | 22.5% | **83.3%** | 10.3% | 6.4% |

achieve SOTA performance by comparing with DreamEdit (Li *et al.*, 2023b) and BLIP-Diffusion (Li *et al.*, 2023a).

### 5.4 Ethics Exploration

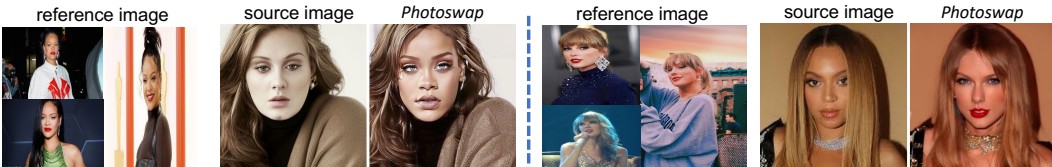

Figure 8: **Results on real human face images across different races**. Evidently, the skin colors are also successfully transferred when swapping a white person with a black person, and vice versa.

Like many AI technologies, text-to-image diffusion models can potentially exhibit biases reflective of those inherent in the training data (Sasha Luccioni *et al.*, 2023; Perera and Patel, 2023). Given that these models are trained on vast text and image datasets, they might inadvertently learn and perpetuate biases, such as stereotypes and prejudices, found within this data. For instance, should the training data contain skewed representations or descriptions of specific demographic groups, the model may produce biased images in response to related prompts.

However, *Photoswap* has been designed to mitigate bias within the generation process of a text-to-image diffusion model. It achieves this by directly substituting the depicted subject with the intended target. In Figure 8, we present our evaluation of face swapping across various skin tones. It is crucial to note that when there is a significant disparity between the source and reference images, the swapping results tend to homogenize the skin color. As a result, we advocate for the use of *Photoswap* on subjects of similar racial backgrounds to achieve more satisfactory and authentic outcomes. Despite these potential disparities, the model ensures the preservation of most of the target subject's specific facial features, reinforcing the credibility and accuracy of the final image.

## 6 Conclusion

This paper introduces *Photoswap*, a novel framework designed for personalized subject swapping in images. To facilitate seamless subject photo swapping, we propose leveraging self-attention control by exchanging intermediate variables within the attention layer between the source image and reference images. Despite its simplicity, our extensive experimentation and evaluations provide compelling evidence for the effectiveness of *Photoswap*. Our framework offers a robust and intuitive solution for subject swapping, enabling users to effortlessly manipulate images according to their preferences. In the future, we plan to further advance the method to address those common failure issues to enhance the overall performance and versatility of personalized subject swapping.

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

## A  Attention Swapping Step Analysis

In this section, we visualize the effect of the influence of swapping steps of different components. As discussed in the main paper, self-attention output $\phi$, and self-attention map $M$, derived from the self-attention layer, encompasses comprehensive content information from the source image, without relying on direct computation with textual features. Previous works such as Hertz *et al.* (2022) did not explore the usage of $\phi$ and $M$ in the object-level image editing process.

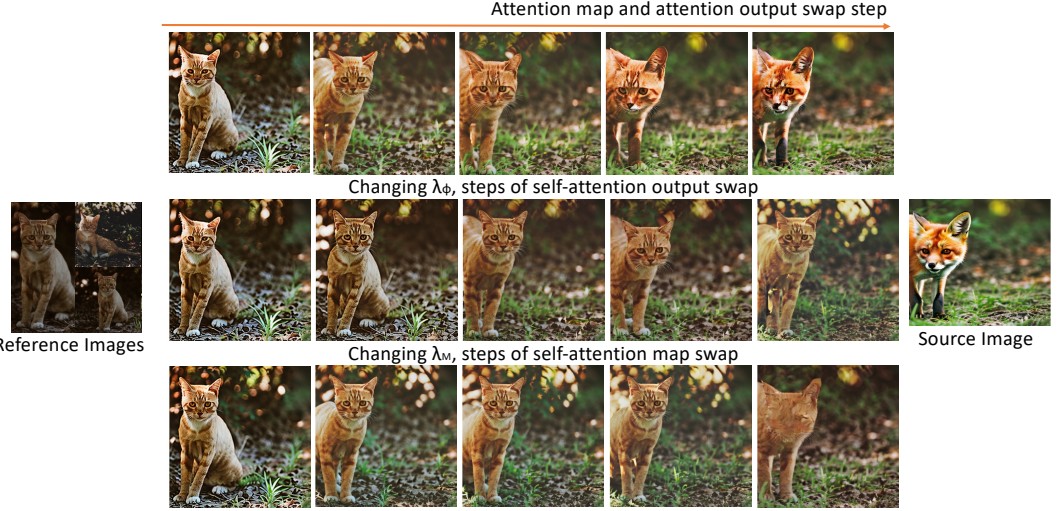

Figure 9: **Results at different swapping steps**. With consistent steps, swapping the self-attention output provides superior control over the layout, including the subject's gestures and the background details. However, excessive swapping could affect the subject's identity, as the new concept introduced through the text prompt might be overshadowed by the swapping of the attention output or attention map. This effect is more clear when swapping the self-attention output $\lambda_\phi$. Furthermore, we observed that replacing the attention map for an extensive number of steps can result in an image with significant noise, possibly due to a compatibility issue between the attention map and the $v$ vector.

Figure 9 provides a visual representation of the effect of incrementally increasing the swapping step for one $\lambda$ hyperparameter while maintaining the other two at zero. Although all of them can be utilized for subject swapping, they demonstrate varying levels of layout control. At the same swapping step, the self-attention output $\phi$ offers more robust layout control, facilitating better alignment of gestures and preservation of background context. In contrast, the self-attention map $M$ and cross-attention map $A$ demonstrate similar capabilities in controlling the layout.

However, extensive swapping can affect the subject's identity, as the novel concept introduced via the text prompt might be eclipsed by the swapping of the attention output or attention map. This effect becomes particularly evident when swapping the self-attention output. This analysis further informs the determination of the default $\lambda_\phi$, $\lambda_M$, and $\lambda_A$ values. While the cross-attention map $A$ facilitates more fine-grained generation control, given its incorporation of information from textual tokens, we discovered that $\phi$ offers stronger holistic generation control, bolstering the overall output's quality and integrity.

## B  Results of Other Concept Learning Methods

We mainly use DreamBooth as the concept learning method in the experiments, primarily due to its superior capabilities in learning subject identities (Ruiz *et al.*, 2023). However, our method is not strictly dependent on any specific concept learning method. In fact, other concept learning methods could be effectively employed to introduce the concept of the target subject.

To illustrate this, we present the results of *Photoswap* when applying Text Inversion (Gal *et al.*, 2023a). We train the model using 8 A100 GPUs with a batch size of 4, a learning rate of 5e-4, and set the training steps to 1000. Results in Figure 10 indicate that Text Inversion also proves to be an effective concept learning method, as it successfully captures key features of the target object.

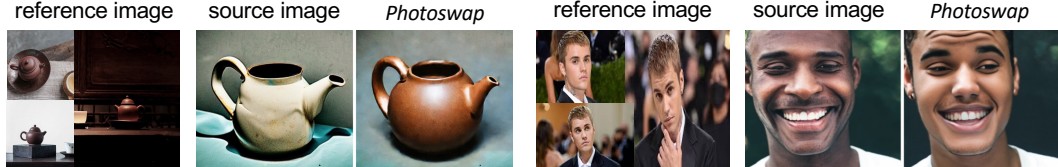

Figure 10: **Results of Text Inversion (Gal *et al.*, 2023a) as the concept learning module**. It can successfully capture key subject features, but its performance drops when representing complex structures such as human faces.

Nevertheless, we observe that Text Inversion performance is notably underwhelming when applied to human faces. We postulate that this is because Text Inversion focuses on learning a new embedding for the novel concept, rather than finetuning the entire model. Consequently, the capacity to express the new concept becomes inherently limited, resulting in its less than optimal performance in certain areas.

## C  Failure Cases

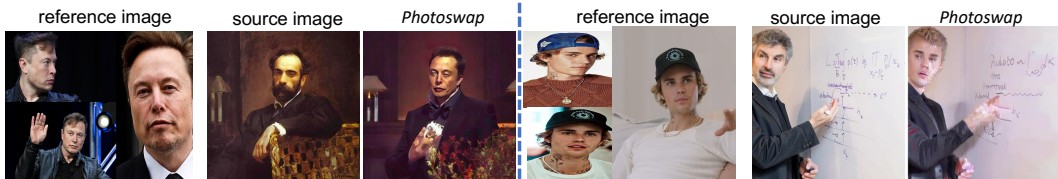

Figure 11: **Failure cases.** The model sometimes struggles to accurately reconstruct hand details and complex background information such as formula on a whiteboard.

Here we highlight two common failure cases. First, the model struggles to accurately reproduce hands. When the subject includes hands and fingers, the swapping results often fail to precisely mirror the original hand gestures or the number of fingers. This issue could be an inherited challenge from Stable Diffusion. Moreover, *Photoswap* can encounter difficulties when the image comprises complex information. As illustrated in the lower row of Figure 11, *Photoswap* fails to reconstruct the complicated formula on a whiteboard. Therefore, while *Photoswap* exhibits strong performance across various scenarios, it's crucial to acknowledge these limitations when considering its application in real-world scenarios involving intricate hand gestures or complex abstract information.

## D  Evaluation Details

For real images, we sourced all our images from internet searches. We employed the search prompt: 'a photo of <target>'. Here, the <target> variable could be a specific celebrity (e.g., 'Elon Musk') or a descriptive scene (e.g., 'a cute yellow cat running in the forest'). The celebrity names were identified through a Google search with the prompt "top celebrities 2023." For scene descriptions, we curated a list of 100 distinct search prompts to source images from the internet. In total, we aggregated 1,000 images using these prompts. All prompts, along with the collected image, will be made available in our next revision.

For synthetic images, we generated 1,000 images using text prompts with the text-to-image diffusion model version 2.1. These prompts spanned a range, including those centered on humans (e.g., "A photo of a woman looking left and smiling, Van Gogh style") and those focused on non-human subject (e.g., "An old car in the middle of the road, flanked by trees during autumn"). All prompts used in synthetic image generation will also be released too. For the human evaluation exhibited in this rebuttal and in the paper, we utilized the "random" package in Python to sample 200 images from both the real and synthetic datasets, respectively. Each image underwent evaluation by five distinct individuals on Amazon Turk. In total, this resulted in a comprehensive 6,000 ratings, as we compared our model against P2P, PnP, and MasaCtrl. Our findings unequivocally indicate that our model surpasses all other methods in performance.

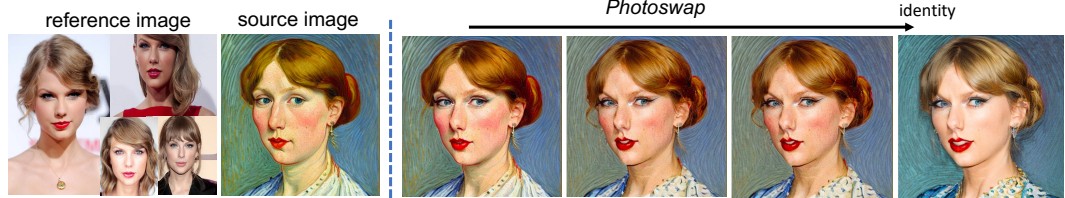

Figure 12: **Ablation results of** $M$ **in** *Photoswap.* With $M$ value increase, the generated one is more similar to the style and identity of source image and dissimilar to the reference subject, and vice versa.

For source image processing, all we do is to resize the image into standard 512x512 pixels. It is worth noting that there is also no postprocessing needed for generated images.

## E    Unet Swapping Layer

There are 16 Unet layers in the Stable Diffusion backbone. We tested the effect of different layers according to their position (up, middle, or lower) in the Unet and their latent size. Through hundreds of experiments on the layer combination, we found that while the latent size of the layer plays a minor role, the position of the layer to be swapped matters. More specifically, we find the essential part is to do swapping operations in all the decoder layers in the Unet.

## F    Controlling Subject Identity

The effectiveness of the proposed mutual self-attention is demonstrated through both synthetic image synthesis and real image editing. Additionally, we perform an analysis of the control strategy with varying values of $M$ during the denoising process. Figure 12 provides insights into this analysis. It is observed that when applying self-attention control with a large swapping step $\lambda_M$ for $M$, the synthesized image closely resembles the source image in terms of both style and identity. In this scenario, all contents from the source image are preserved, while the subject style learned from the reference subject is disregarded. As the value of $M$ decreases, the synthesized image maintains the subject from the reference image while retaining the layout and pose of the contents from the source image. This gradual transition in the control strategy allows for a balance between subject style transfer and preservation of the original image's contents.

