# OpenReview forum: "PHOTOSWAP: Personalized Subject Swapping in Images"
_NeurIPS.cc/2023/Conference — NeurIPS 2023 poster_

### Official Review · Reviewer_Ep7B · 2023-07-04

**Soundness:** 2 fair
**Presentation:** 2 fair
**Contribution:** 2 fair
**Rating:** 5
**Confidence:** 3

**Summary:**

This paper presents a combination of DDIM inversion and Dreambooth to customize the appearance of existing images.

The authors also experimented with some attention layer hacking.

**Strengths:**

1.	Paper is easy to read.
2.	Framework is reasonable. Dreambooth+DDIM inversion+Attention hacking will really lead to such results.
3.	Easy to reproduce.


**Weaknesses:**

1.	My main concern is the novelty: it seems that everything of in the combo of “Dreambooth+DDIM inversion+Attention hacking” are proposed and/or extensively discussed in previous works.  However, we are also aware of that many recent papers in this direction (like masactrl) are non-peer-reviewed works so that I am not leaning negative because of the novelty. Nevertheless, the combo of these components still looks a bit ad-hoc to me.
2.	The experiments need some improvements: reference guided diffusion is sensitive to inputs. We should present random non-cherry-picked samples to study the method performance.


**Questions:**

Why we mainly conduct the comparison against prompt-to-prompt? Shouldn’t we mainly consider frameworks like imagic/masactrl to edit real image?

**Limitations:**

see Weaknesses

---

> ### Author Rebuttal · Authors · 2023-08-09
>
> Thanks for the suggestions, we will explain the concerns as follows:
>
> > About the paper novelty.
>
> a.  Personalized subject swapping is an emerging vision task that has abundant user applications in practice.  The task we undertook is inherently challenging, due to the lack of training pairs. In the literature, there are few works that can address the problem in a unified way.    In contrast, as the results are shown in the framework, the proposed method is robust and can be generalized to different domains including human faces, animals, daily objects, vector art, and paintings.
>
> b.  We proposed a unified framework to address the problem.  The attention map manipulation process is non-trivial.  In section 4, all the steps in our method are based the observation of attention map analysis.  Without Photoswap,  simply applying existing attention map manipulation methods will not works well, as they could not preserve the non-swapped region well.   We use DreamBooth and DDIM inversion for implementation simplicity. Other concept learning (Lora, Suti) methods and inversion methods will also be compatible with our Photoswap framework.
>
>
> > Why not compare to Masactrl?
>
> MasaCtrl is a non-peer-reviewed work, and it is released on arXiv just a month before the NeurIPS submission. But we still cited and discussed it when we noticed the paper before submission. Our paper is targeting a fundamentally different task, personalized subject swapping, while their method is for subject gesture changing.
>
> Nonetheless, we have presented a comparison between Photoswap and MasaCtrl during this rebuttal stage, which is shown in Table 1 in the attached PDF and will be incorporated into our next version. We reproduced the results from MasaCtrl and acknowledged its commendable performance in altering subject gestures. However, it has notably inferior outcomes in the personalized subject swapping task. Through abundant human evaluation, Photoswap outperforms MasaCtrl across all metrics.
> Upon closer inspection, we observed that MasaCtrl primarily swaps the q vector in self-attention, having determined a strong correlation between the q vector and the subject shape. In our targeted task, the complete swapping of the source subject to the target subject necessitates a more comprehensive approach. Both the attention map—derived from the interaction of q and k—and the attention output—resulting from q, k, and v—is crucial.
>
> | Metrics      | Photoswap    | MasaCtrl+DreamBooth   | Tie |
> | :----        |    :-----:   |          :-----:|   ----:|
> | Subject Identity Preservation      |**79.1%**| 10.3%   | 10.6% |
> | Background Preservation   |     **72.8%**   | 10.2%      |  17.0%|
> | Overall Swapping Quality   |   **83.3%**     | 10.3%| 6.4%   |
>
> > A combo of methods
>
> Personalized subject swapping is an interesting and challenging task, with our logical framework, Photoswap, we successfully addressed this complex challenge. It's important to emphasize that Photoswap is a versatile framework that can integrate various methodological elements. For instance, one can seamlessly incorporate LoRA weights into Photoswap or introduce a controlnet layer for more refined swapping. The task we embarked upon is fundamentally challenging, but we secured significant outcomes. It's essential to emphasize that a direct combination of DreamBooth with the earlier proposed attention map insertion is not optimal, as clearly demonstrated in the paper's P2P comparison and the rebuttal's P2P comparisons. Unlike prior approaches, we performed attention swapping on a broader set of attention variables. We also delved deeper into the efficacy of these variables, offering valuable insights for future investigations.
>
>
> > Improvements for experiments.
>
> As is exhibited in the paper, Photoswap is a subject swapping model for general images, including real images and synthetic images, the target ranges from human faces to common house items, from daily life images, movie pictures, to artistic works. During the human evaluation, we did 2000 comparisons on both real images and synthetic images. All the swapping results used for evaluation are non-cherry pick results. We will also provide code for performance reproduction.

---

> > ### Author Response · Authors · 2023-08-15
> > **Thanks for your insights. Do you have any other questions?**
> >
> > First and foremost, we'd like to express our sincere gratitude for your constructive and encouraging feedback on our paper. We provided further clarification on Photoswap and the comparison result with MasaCtrl is attached in the rebuttal.
> >
> > Having made efforts to address your insights, we'd like to ensure that our explanations of Photoswap and additional experiments are both comprehensive and satisfying. If there are any lingering questions, please don't hesitate to bring them to our attention. Your continued guidance will only help in enhancing the quality and clarity of our work.
> >
> > Thank you once more for your invaluable feedback and support.

---

> ### Comment · Reviewer_Ep7B · 2023-08-21
>
> Thanks for the author response. This reviewer kill keep the original rate after considering the additional materials.

---

> > ### Author Response · Authors · 2023-08-21
> >
> > Thanks so much for your time and efforts in reviewing our paper!

---

### Official Review · Reviewer_p4oH · 2023-07-06

**Soundness:** 2 fair
**Presentation:** 1 poor
**Contribution:** 2 fair
**Rating:** 3
**Confidence:** 4

**Summary:**

This paper introduces a new method for inserting a subject into a target image. The approach consists of 1) using dreambooth to extract the appearance information of the subject, and 2) copy the attention from the target image (obtained by regenerate the target image using DDIM inversion) to control the layout for the generated image. Empirical results show that the proposed method is preferred by raters ~50% of the time compared with P2P baseline.

**Strengths:**

The proposed method is straightforward, and the empirical results look promising.

**Weaknesses:**

* Technical contribution not clear. As mentioned in the paper, the two main components used in the paper, i.e. dreambooth and attention copying, have been proposed in prior works. It is unclear what the main technical contribution of this work is. In fact, it is even not clear, based on the limited description, what the main difference between the proposed method and the baseline is.


* Unclear presentation. The approach section is hard to follow. In particular, Sec. 4.1 and 4.2 are distracting and it is unclear what information these two sections try to convey. The experiment section does not provide sufficient information for reproducing and understanding the implications of the experiments, e.g. what are the data, how were the baseline implemented, what the data and instructions are for the user study?


* Limited experiments. While the authors claim that one of the main contributions of this work is extensive experiments, the actual evaluation is very limited. For a reasonable research paper, I will expect more solid and extensive evaluation regarding different aspects of the proposed approach, including but not limited to the success rate of the proposed approach, the effect of target image content and subject image, the effect of meta parameters, etc. From the experiments in the paper, it is even unclear how applicable the proposed method is in practice.


**Questions:**

Please refer to the weakness section.

**Limitations:**

This paper does not discuss the limitation. While there's a section for ethical issues, the section doesn't provide useful discussion.

---

> ### Author Rebuttal · Authors · 2023-08-10
>
> Thanks to the reviewer for your insightful suggestions. Here are our response to your concerns.
>
> > What is technical contribution. What is the main difference between the proposed method and the baseline.
>
> Quote Reviewer z7Bi, “The task of personalized subject swapping in images is fancy and interesting. This is the first work that could handle such challenging swapping task.” Such a task requires seamlessly integrating a new subjects into existing images in the exect position of the source subject. While previous methods are usually restricted to global editing, we achieved great results with logical framework, Photoswap. Note that Photoswap is a framework, which could incorporate more methods and components. For example, one could easily add LoRA weights into Photoswap or add controlnet layer to achieve fine-grained swapping.
> The primary distinction between our approach and the baseline centers around the attention swapping process. In P2P, the authors predominantly use the cross-attention map for image editing, having identified a strong correlation between this map and the resultant image. In contrast, our technique employs attention swapping on the cross-attention map, self-attention map, and self-attention output, allowing for more detailed adjustments. As showcased in Figure 7 of our paper and the attached pdf file, PhotoSwap excels in both subject swapping and background conservation. The human evaluation results in Table 1 show that Photoswap outperforms the baseline by a large margin.
>
> > Unclear presentation in Section 4
>
> In Section 4, we present the framework of Photoswap. Photoswap first learns a concept image into a token by a concept learning method, as in section 4.1. The training-free attention swapping process further keeps the background information unchanged during the target image generation process, and the learned subject is injected into the image through textual prompt as in section 4.2. Please refer to Algorithm 1 in the paper for a detailed process. We will also release the code for performance reproduction.
>
> > What are the data? How was the baseline implemented? What the data and instructions are for the user study?
>
> In terms of concept source, for non-human, we utilized the images from the DreamBooth benchmark as our concept learning method is based on DreamBooth. For human, we collect celebrity images from the Internet. All the source images are also from the Internet. For the baseline, we use DreamBooth to transform a new concept into a textual token and then employ P2P as the attention-based process for subject swapping. The data used for the user study was swapping results on randomly sampled synthetic images and real images from the Internet. The user instructions and our user study interface are also attached in the pdf.
> We will add those details to the next version. Please let us know if there are any more details missing in your mind.
>
> > Limited experiments
>
> In the paper, we have tested Photoswap widely from synthetic images to real images, from human images to non-human subjects. We conducted experiments on complicated situations such as multiple subjects and occluded subjects. Since this is a new task, we built a baseline and did both human evaluation and qualitative comparison. We also conducted ablation study on the effect of different hyperparameters on human faces and common subjects. To validate the generalization of our framework, we also test that on other concept learning methods. We also did experiments to test the potential ethical concerns. Lastly, we also discussed the failure cases. In this rebuttal period, we further conducted a large-scale human evaluation on more baselines.
>
> > Success rate. Effect of target image content and subject image. Effect of meta parameters. From the How applicable the proposed method is in practice.
>
> In our study, we assessed Photoswap's capability for subject swapping within images, a challenge that had not been effectively addressed before. Whether transitioning from real to synthetic images, everyday subjects to human faces, or artistic creations to film stills, we have illustrated Photoswap's proficiency in seamlessly substituting a subject from one image with a personalized subject from another. Both in the original paper and throughout this rebuttal, we've noted a commendably high success rate (exceeding 50%) in terms of users identifying the context from the source image and recognizing the identity from the reference image. The effects of the three hyperparameters, which influence the swapping steps, are detailed in the appendix. We will make our source code available for further examination of Photoswap's applicability.
> We would be happy to address them if the reviewer has more specific suggestions or concerns during the discussion phase.

---

> > ### Author Response · Authors · 2023-08-15
> > **Thanks for your valuable feedback. Do you have other questions?**
> >
> > We deeply appreciate the time and effort you've invested in reviewing our manuscript. We have done our best to address each of your concerns in the preceding responses. To ensure our revisions and explanations meet your expectations, could you please let us know if there are any additional questions or if any areas remain unresolved in your view? We strive for clarity and thoroughness and want to ensure that we've attended to all your points adequately.
> >
> > Thank you once again for your invaluable insights and feedback.

---

> ### Comment · Reviewer_p4oH · 2023-08-20
>
> Thanks the authors for the response. While the rebuttal partially address my concern, most of them lack concrete information and are not sufficient to address  the concern.
>
> 1. If the contribution and performance gain comes from a better attention swapping mechanism, I would expect the a more thorough description and a solid ablation study to verify the claim. The current paper does not provide sufficient information for the improvements made by this work and their contribution.
>
> 2. The concerns regarding the presentation are not resolved. Note that releasing the code does not help unless the code is also reviewed for completeness and clarity.
>
> 3. While I understand it is hard to "reproduce" the data collection process, "collecting from internet", "randomly sampled" are still to vague. Information regarding the data source, e.g. how they are crawled, the size, any selection process, etc. should be provided.
>
> 4. It would be more informative if the authors can provide how the success rate is measured and what's the criteria of success. Also, a few qualitative examples in the paper are not sufficient to be considered as a meaningful ablation study. To make it a meaningful study, the authors need to show how consistent the results are.
>
> In summary, similar to the original paper, the claims in the rebuttal are reasonable but are not well supported by concrete evidence.

---

> > ### Author Response · Authors · 2023-08-21
> > **Response (1/2)**
> >
> > Thanks for your feedback on our rebuttal. Here we would like to further resolve your concerns.
> >
> > >If the contribution and performance gain comes from a better attention swapping mechanism, I would expect a more thorough description and a solid ablation study to verify the claim. The current paper does not provide sufficient information on the improvements made by this work and their contribution.
> >
> >
> > Photoswap pioneers a novel challenge and consistently delivers superior performance across different image domains. To ensure an objective comparison and demonstrate the effectiveness of our attention swapping process, we established baselines based on P2P, PnP, and MasaCtrl, which also utilize attention-based mechanisms to achieve image editing. As is shown in the rebuttal pdf and text, Photoswap outperforms all other methods by a large margin, which shows the superiority of our training-free attention swapping process. The results are also attached below.
> >
> > | Metrics                       | Ours  | P2P+DreamBooth | Tie   |
> > |-------------------------------|-------|----------------|-------|
> > | Subject Identity Preservation | 0.434 | 0.300          | 0.266 |
> > | Background Preservation       | 0.393 | 0.302          | 0.305 |
> > | Overall Swapping Quality      | 0.373 | 0.271          | 0.356 |
> >
> > Table 1. Human evaluation comparison with P2P+DreamBooth.
> >
> > | Metrics                       | Ours  | PnP+DreamBooth | Tie   |
> > |-------------------------------|-------|----------------|-------|
> > | Subject Identity Preservation | 0.527 | 0.221          | 0.252 |
> > | Background Preservation       | 0.491 | 0.207          | 0.302 |
> > | Overall Swapping Quality      | 0.551 | 0.224          | 0.225 |
> >
> > Table 2. Human evaluation comparison with PnP+DreamBooth.
> >
> > | Metrics                       | Ours  | MasaCtrl+DreamBooth | Tie   |
> > |-------------------------------|-------|---------------------|-------|
> > | Subject Identity Preservation | 0.791 | 0.103               | 0.106 |
> > | Background Preservation       | 0.728 | 0.102               | 0.170 |
> > | Overall Swapping Quality      | 0.833 | 0.103               | 0.064 |
> >
> > Table 3. Human evaluation comparison with MasaCtrl+DreamBooth.
> >
> > In contrast to the image classification domain where a ground truth typically exists, the image subject swapping task we address is novel and lacks an established ground truth or evaluative benchmark. Current image editing works including P2P[1], PnP[2], and MasaCtrl[3] mainly rely on human evaluation to prove efficacy. As highlighted by Reviewer TtiF, "Considering the lack of unified metrics for this task, human evaluation is adequate". Comprehensive human evaluations and qualitative analyses indicate that our model significantly surpasses other methods in performance.
> >
> > Although human evaluation already provides the most direct evaluation for model performance, we also employed the DINO and CLIP-I metrics to measure image similarity following DreamBooth [4]. First, we assess subject identification preservation by examining the similarity between the generated image and the concept image. Subsequently, background preservation is evaluated by determining the similarity between the generated image and its source counterpart. From the data presented in the table, it is clear that Photoswap consistently outperforms competing methods across both evaluated metrics. This aligns with the findings from our human evaluation, in which Photoswap surpassed other methods in all metrics, including both Subject Identity Preservation and Background Preservation.
> >
> > |        | Ours | P2P+DreamBooth | PnP+DreamBooth | MasaCtrl+DreamBooth |
> > |--------|------|----------------|----------------|---------------------|
> > | DINO   | 0.55 | 0.44           | 0.42           | 0.31                |
> > | CLIP-I | 0.80 | 0.72           | 0.68           | 0.53                |
> >
> > Table 4. Automatic evaluation on subject identity.
> >
> > |        | Ours | P2P+DreamBooth | PnP+DreamBooth | MasaCtrl+DreamBooth |
> > |--------|------|----------------|----------------|---------------------|
> > | DINO   | 0.78 | 0.72           | 0.73           | 0.70                |
> > | CLIP-I | 0.89 | 0.79           | 0.76           | 0.69                |
> >
> > Table 5. Automatic evaluation on background information preservation.
> >
> > We also further tested the effectiveness of all the swapping variables used in this paper. In Figure 8 and Section 5.3, the impact of the self-attention map M is elucidated. Further insights on the significance of the overall attention map variable and the attention output variable in the subject swapping process are presented in Figure 5 and Appendix Section C. This section also delves into the influence of each swapping phase.

---

> > > ### Author Response · Authors · 2023-08-21
> > > **Response (2/2)**
> > >
> > > >The concerns regarding the presentation are not resolved. Note that releasing the code does not help unless the code is also reviewed for completeness and clarity.
> > >
> > > Could you provide further insight into the concerns surrounding the presentation? Within Section 4, we delineate our primary methodology. To delve deeper, Section 4.1 elucidates our approach to incorporating a novel concept into the model, which subsequently facilitates its use in text prompts. Subsequently, in Section 4.2, we expound upon the training-free attention swapping procedure. This comprehensive process is also visually represented in Algorithm 1. Should there remain any ambiguities, we are more than willing to offer additional clarifications. We deeply appreciate your feedback and suggestions.
> > >
> > > >While I understand it is hard to "reproduce" the data collection process,   "collecting from internet", "randomly sampled" are still to vague. Information regarding the data source, e.g. how they are crawled, the size, any selection process, etc. should be provided.
> > >
> > > We acknowledge that our initial description of the source image collection process was not sufficiently clear. In this context, we provide an in-depth explanation.
> > >
> > > For real images, we sourced all our images from internet searches. We employed the search prompt: 'a photo of <target>'. Here, the <target> variable could be a specific celebrity (e.g., 'Elon Musk') or a descriptive scene (e.g., 'a cute yellow cat running in the forest'). The celebrity names were identified through a Google search with the prompt "top celebrities 2023." For scene descriptions, we curated a list of 100 distinct search prompts to source images from the internet. In total, we aggregated 1,000 images using these prompts. All prompts, along with the collected image, will be made available in our next revision.
> > >
> > > For synthetic images, we generated 1,000 images using text prompts with the text-to-image diffusion model version 2.1. These prompts spanned a range, including those centered on humans (e.g., "A photo of a woman looking left and smiling, Van Gogh style") and those focus on non-human subject (e.g., "An old car in the middle of the road, flanked by trees during autumn"). All prompts used in synthetic image generation will also be released too. For the human evaluation exhibited in this rebuttal and in the paper, we utilized the "random" package in Python to sample 200 images from both the real and synthetic datasets, respectively. Each image underwent evaluation by five distinct individuals on Amazon Turk. In total, this resulted in a comprehensive 6,000 ratings, as we compared our model against P2P, PnP, and MasaCtrl. Our findings unequivocally indicate that our model surpasses all other methods in performance.
> > >
> > > For source image processing, all we do is to resize the image into standard 512x512 pixels. It is worth noting that there is also no postprocessing needed for generated images.
> > >
> > >
> > > >It would be more informative if the authors can provide how the success rate is measured and what's the criteria for success. Also, a few qualitative examples in the paper are not sufficient to be considered a meaningful ablation study. To make it a meaningful study, the authors need to show how consistent the results are.
> > >
> > >
> > > To the best of our knowledge, there is no existing automatic evaluation metric for success rate on image editing. Due to the lack of ground truth and standard evaluation metrics, current image editing works such as P2P[1], PnP[2], and MasaCtrl[3] are mostly using human evaluation and qualitative comparison to demonstrate performance.
> > >
> > > In response to your feedback, we expanded our analysis to incorporate human evaluations focused on the success rate.
> > > With respect to quantifying the success rate, we established three criteria to define a successful image swap:
> > > * The generated image featuring the replaced subject should exhibit high quality.
> > > * The background details not pertaining to the targeted subject should remain consistent with the original source image.
> > > * The outcome of the swap should be recognizable as the subject from the concept image, mirroring the pose, gesture, and facial expression of the original subject.
> > >
> > > Presented below are our findings, based on a sample of 200 real images juxtaposed with 200 synthetic images:
> > >
> > > |                   | Ours | P2P+DreamBooth | PnP+DreamBooth | MasaCtrl+DreamBooth |
> > > |-------------------|------|----------------|----------------|---------------------|
> > > | Human subject     | 0.57 | 0.37           | 0.34           | 0.12                |
> > > | Non-human subject | 0.72 | 0.51           | 0.43           | 0.14                |
> > >
> > > Table 6. Human evaluation results on Success Rate.
> > >
> > > In the table, results from the human evaluator our method outperforms other methods in terms of success rate. Thanks for the insightful suggestions on the success rate, and we will include this in the next revision.

---

> > > > ### Author Response · Authors · 2023-08-21
> > > > **Paper cited in previous comments**
> > > >
> > > > [1] Hertz, Amir, et al. "Prompt-to-Prompt Image Editing with Cross-Attention Control." The Eleventh International Conference on Learning Representations. 2022.
> > > >
> > > > [2] Tumanyan, Narek, et al. "Plug-and-play diffusion features for text-driven image-to-image translation." Proceedings of the IEEE/CVF Conference on Computer Vision and Pattern Recognition. 2023.
> > > >
> > > > [3] Cao, Mingdeng, et al. "MasaCtrl: Tuning-Free Mutual Self-Attention Control for Consistent Image Synthesis and Editing." arXiv preprint arXiv:2304.08465 (2023).
> > > >
> > > > [4] Ruiz, Nataniel, et al. "Dreambooth: Fine tuning text-to-image diffusion models for subject-driven generation." Proceedings of the IEEE/CVF Conference on Computer Vision and Pattern Recognition. 2023.

---

### Official Review · Reviewer_TtiF · 2023-07-07

**Soundness:** 3 good
**Presentation:** 3 good
**Contribution:** 2 fair
**Rating:** 7
**Confidence:** 3

**Summary:**

The authors present a solution to the problem of personalized subject swapping, where the goal is to replace the subject in an image with another user-defined subject. Authors leverage pre-trained diffusion models to make local edits to an input image based on a collection of images of the subject to be inserted. This is done through concept inversion on images of a target subject and scheduling attention map swapping during the diffusion process, switching between the use of attention maps from the original diffusion process and the newer process conditioned on the concept modified text prompt.

**Strengths:**

- Clarity: Work is well presented and method is logically sound. Motivation behind method is also well justified. For example, the attention map visualizations are helpful in motivating their scheduled attention swapping method.
- Significance: The problem of personalized subject swapping is practically interesting and hasn't been explored with depth in prior works. The practical effectiveness of large image diffusion models have been limited by its controllability. This work takes a reasonable step towards making such models accessible and useful to users who don't have domain specific knowledge in image editing.
- Quality of results: Experimental results are impressive, and achieving subject swapping with high fidelity.


**Weaknesses:**

- Missing baseline: Recent work such as Plug-and-Play diffusion has also explored the effectiveness of feature and attention map injection during the diffusion process for image editing. Using DreamBooth + plug-and-play [1] is a more reasonable comparison than using P2P. It is also an important comparison as it is not clear to me whether the author's proposed attention map swapping method is really more effective than the method proposed int plug-and-play.
- Evaluation: Considering the lack of unified metrics for this task, human evaluation is adequate. However, it would be helpful to see other metrics such as subject fidelity [2] reported. It is unclear whether this method leads to lower subject preservation than prior work like DreamBooth.
- Results: Results shown all include subject swapping that are very similar in nature (e.g. cars are swapped for a different looking car). Subjects being swapped may not necessarily be similar in nature (see questions sections for more detailed comment).
- Contribution: both concept inversion and editing through attention map insertion have been explored in prior work, this work seems to combine both effectively, but contribution from their method is relatively small.


[1] Tumanyan, N., Geyer, M., Bagon, S., & Dekel, T. (2022). Plug-and-Play Diffusion Features for Text-Driven Image-to-Image Translation. ArXiv, abs/2211.12572.
[2] Ruiz, N., Li, Y., Jampani, V., Pritch, Y., Rubinstein, M., & Aberman, K. (2022). DreamBooth: Fine Tuning Text-to-Image Diffusion Models for Subject-Driven Generation. ArXiv, abs/2208.12242.

**Questions:**

- How does this method perform when there is a large domain shift in the edited subject? For example, if I wanted to replace a car with a tree, does the method still perform well?
- Does the structure of the subject in the reference image fix the structure of the inserted subject?

**Limitations:**

Authors adequately address the limitations and potential ethical risks in using image diffusion models trained on internet-scale data, especially discussions on the potential biases present in these models.

---

> ### Author Rebuttal · Authors · 2023-08-10
>
> Thanks reviewer for the appreciation on our motivation, paper writing, technical contribution and results.   We will explain also the concerns as follows:
>
> > Comparison with PnP+Dreambooth.
>
> We have attached a comprehensive human evaluation both here and in Table 1 from the pdf. We could see that Photoswap outperforms PnP by a large margin across all metrics.
> PnP was designed for changing the image style, where background details preservation is not the focus. PnP utilizes the text prompt to inject the style information while copying intermediate variables in the self-attention layer to keep the layout unchanged. However, in our task, the background is supposed to be the same while the source subject should be swapped into the target subject. To keep the background unchanged, the only difference between the source prompt and the target prompt in Photoswap is the subject name. Our training-free mechanism is also designed for keeping the background details while swapping the target.
>
> | Metrics      | Photoswap    | PnP+DreamBooth   | Tie |
> | :----        |    :-----:   |          :-----:|   ----:|
> | Subject Identity Preservation      |**52.7%**| 22.1%   | 25.2% |
> | Background Preservation   |     **49.1%**   | 20.7%      |  30.2%|
> | Overall Swapping Quality   |   **55.1%**     | 22.4%| 22.5%   |
>
> > Evaluation metric could be improved:
>
> We shared the same opinion that human evaluation is important since this is a new task. We have conducted a larger-scale human evaluation and the results are attached in the pdf. We also believe subject preservation is important, and one metric used for human evaluation is subject identity recognition. Our Photoswap outperforms all baselines by a large margin. We also plan to add more metrics for evaluation given more time for the revision.
>
> > Could it swap between dissimilar subjects?
>
> Yes. Photoswap does not need explicit mask of the source subject. The attention swapping in the Unet layer could be between any two shapes of subjects.
>
>
> > Contributions
>
> As you and Reviewer z7Bi mentioned, The task we undertook is inherently challenging, yet we have managed to achieve noteworthy results. Besides, we want to highlight that our simple training-free structure achieved amazing results without complicated structure. Our method could easily combined with other structure such as LoRA or ControlNet. As is well illustrated in the P2P comparison in the paper, and P2P, MasaCtrl, PnP comparison in this rebuttal, our model achieves a much better performance than all baselines across all metrics. We further analyzed the effectiveness of attention map and attention output, which provide insight for training-free image editing work.
>
>
> > How does the method perform when swapping has large domain gap?
>
> Fundamentally, our method does not require similarity between source domain and target domain since we directly swap the attention variables. In Figure 8 in the paper, we show it works when swapping between portrait and painting.
>
> > Does the structure of the subject in the reference image fix the structure of the inserted subject?
>
> No, the structure of the subject is not rigidly defined. Once the identity of the subject is learnt, it adapts to the gesture of the source subject. As exemplified in the teaser image of our paper, the human faces or subjects in the resultant images align with the source image, instead of merely duplicating the reference image.

---

> > ### Author Response · Authors · 2023-08-15
> > **Thanks for your suggestions. Do you have any other questions?**
> >
> > We're truly grateful for your encouraging words on our work. We've taken care to address your comments and provide additional clarifications where needed. The PnP baseline comparison is also attached in the rebuttal.
> >
> > To ensure that our responses have fully addressed any concerns, we kindly ask if you have any further questions or if there are specific areas where you believe more clarification might be beneficial. We value your expertise in image generation/editing and aim to ensure that our paper is as clear and comprehensive as possible.
> >
> > Once again, thank you for your supportive and constructive insights.

---

> ### Comment · Reviewer_TtiF · 2023-08-18
> **Raised score to accept**
>
> My initial concerns were mostly on the evaluation method, degree of contribution and performance of method in edge cases. I agree with the authors that given the nature of the problem, human evaluation is sufficient. I am still on the fence about the novelty of the method, but given the impressive results I am willing to raise my initial score.

---

> > ### Author Response · Authors · 2023-08-18
> > **Thank you for acknowledging our work**
> >
> > We're glad that our human evaluation on this challenging task provided sufficient support for Photoswap. We have taken note of your additional feedback and will incorporate the suggested changes to further improve the paper. and we hope our work will make a valuable contribution to image editing/image generation. Again, we sincerely appreciate your positive feedback and acknowledgment of the results presented in our paper.

---

> > > ### Author Response · Authors · 2023-08-21
> > > **Additional experimental results.**
> > >
> > > We did more experiments on subject fidelity as you previously mentioned in DreamBooth [1]. Following DreamBooth, we also evaluated DINO and CLIP-I. The results show that our model consistently outperforms all other methods, aligning with the findings from human evaluations.
> > >
> > > |        | Ours | P2P+DreamBooth | PnP+DreamBooth | MasaCtrl+DreamBooth |
> > > |--------|------|----------------|----------------|---------------------|
> > > | DINO   | 0.55 | 0.44           | 0.42           | 0.31                |
> > > | CLIP-I | 0.80 | 0.72           | 0.68           | 0.53                |
> > >
> > > Table 1. Automatic evaluation on subject identity.
> > >
> > > |        | Ours | P2P+DreamBooth | PnP+DreamBooth | MasaCtrl+DreamBooth |
> > > |--------|------|----------------|----------------|---------------------|
> > > | DINO   | 0.78 | 0.72           | 0.73           | 0.70                |
> > > | CLIP-I | 0.89 | 0.79           | 0.76           | 0.69                |
> > >
> > > Table 2. Automatic evaluation on background information preservation.
> > >
> > > [1] Ruiz, Nataniel, et al. "Dreambooth: Fine tuning text-to-image diffusion models for subject-driven generation." Proceedings of the IEEE/CVF Conference on Computer Vision and Pattern Recognition. 2023.

---

### Official Review · Reviewer_z7Bi · 2023-07-07

**Soundness:** 3 good
**Presentation:** 3 good
**Contribution:** 3 good
**Rating:** 5
**Confidence:** 4

**Summary:**

In this paper, they propose Photoswap, which could seamlessly swap personalized subjects into source images. The swap process is training-free and only leverages the manipulation of self-attention and cross-attention. The swapped object could maintain the pose of the source image without hurting the coherence of the image. The results are promising on both synthetic images and real images.

**Strengths:**

- The task of personalized subject swapping in images is fancy and interesting. This is the first work that could handle such challenging swapping task.
- The paper excels in providing thorough illustrations to explain the design approach, making it easier for readers to understand the intricacies of the proposed method. Furthermore, the clear determination of parameters is commendable, as it enhances the reproducibility and applicability of the research.
- The results presented in the paper are impressive. Even when dealing with challenging scenarios such as multi-subject swap and occluded subject swap, the proposed method performs remarkably well.

**Weaknesses:**

- In the user study, the comparisons are conducted on 99 examples. Although the number of test images and total votes (3 votes per example) is small, it may introduce bias into the results. I recommend addressing this concern by either increasing the number of test images. Additionally, it is unclear from the paper whether the user study was conducted solely on synthetic images. It would be valuable to explore the performance of your proposed method on real images as well, as this would provide further insights into its practical applicability and generalization.
- In Fig. 7, the presented results are not convincingly better than those of the P2P+dreambooth method. To strengthen your claims, I suggest providing additional comparisons, especially on real images.
- When discussing the swapping of the self-attention layer, it is unclear which self-attention layer in the U-Net architecture you are referring to. Please provide clarification on which specific self-attention layer is being swapped, as this information is essential for replicating your results and understanding the significance of this modification.

**Questions:**

refer to Weaknesses

**Limitations:**

Authors have stated the limitations.

---

> ### Author Rebuttal · Authors · 2023-08-10
>
> Thanks for the reviewer on the acknowledge of our task and appreciation of the results. Besides the great suggestion,  we will explain all the concerns as follows:
>
> > larger scale human evaluation comparison with P2P on both synthetic and real images
>
> Thanks so much for the suggestion. During this rebuttal, we conducted more human evaluation on 400 image pairs containing 200 real images and 200 synthetic images, with each having 5 voting. Here, as you suggested we share the P2P+dreambooth baseline comparision below, where Photoswap consistently outperforms P2P on all three aspects.
>
> | Metrics      | Photoswap    | P2P+DreamBooth   | Tie |
> | :----        |    :-----:   |          :-----:|   ----:|
> | Subject Identity Preservation      |**43.4%**| 30.0%   | 27.6% |
> | Background Preservation   |     **39.3%**   | 30.2%      |  30.5%|
> | Overall Swapping Quality   |   **37.3%**     | 27.1%| 35.6%   |
> Table 1.  Human evaluation on  400 images with 200 real image 200 synthetic images. Each image pair contains 5 ratings from Amazon Turk.
>
> > Qualitative comparison with P2P on real image.
>
> As suggested, we attached the qualitative comparison with P2P+DreamBooth in Figure 2 in the attached pdf. From the example, we can clearly see that Photoswap performs better on both subject swapping and background preservation, especially on preserving the pose of the source subject.  We will include more comparisons on real image in the revision.
>
> > Please provide clarification on which specific self-attention layer is being swapped
>
> There are 16 Unet layers in the Stable Diffusion backbone. We tested the effect of different layers according to their position (up, middle, or lower) in the Unet and their latent size. Through around hundreds of experiments on the layer combination, we found that while the latent size of the layer play a minor role, the position of the layer to be swapped matters. More specifically, we find the essential part is to do swapping operation in all the decoder layers in the Unet. We will also release code to verify all findings to ensure all the results can be reproducible.

---

> > ### Author Response · Authors · 2023-08-15
> > **Thanks for your kind words. Do you have other suggestions?**
> >
> > Thank you for your acknowledgment and valuable feedback for Photoswap. We have addressed each of your concerns in the rebuttal above. We would like to ensure that our responses have provided clarity and satisfactorily addressed your queries.
> >
> > Should you have any further questions, or if there are areas where you feel my response might benefit from additional clarification, please do not hesitate to let us know. It is of utmost importance to me to ensure that all your concerns are fully addressed.
> >
> > We truly appreciate your support and guidance in this process.

---

### Author Rebuttal · Authors · 2023-08-10

Attached pdf

---

### Author Response · Authors · 2023-08-20
**Request for feedback on Rebuttal for submission #5128**

Dear Reviewers,

I hope this message finds you well. Firstly, I'd like to extend my sincere gratitude for the time and effort you've dedicated to reviewing our paper. Your insights have been invaluable to us.

We've submitted our rebuttal and understand that everyone has a busy schedule. We kindly wanted to remind and request those who haven't yet had a chance to review our responses to please provide their feedback. We also want to thank reviewer TtiF for feedback on our rebuttal.

Since the Author-Reviewer discussion phase ends on Aug 21st, it would be extremely helpful for us to receive your final thoughts, as it would guide our subsequent revisions and contribute to the betterment of our work.

Thank you once again for your expertise and dedication to the peer review process. We truly appreciate it.

Warm regards,

Authors of submission #5128

---

### Decision · Program_Chairs · 2023-09-21

**Decision:**

Accept (poster)

**Comment:**

Reviewers and the AC read the rebuttal and took that into consideration for their final recommendation. The AC believes that the answers to the unresponsive reviewer in the rebuttal reasonably addressed their concerns. Many suggestions for improving the exposition, the baseline comparisons, and the evaluations were discussed with reviewers in the rebuttal. They should be included in the camera-ready paper.